# RETHINKING DEEP ACTIVE LEARNING:
# USING UNLABELED DATA AT MODEL TRAINING

## ABSTRACT

Active learning typically focuses on training a model on few labeled examples alone, while unlabeled ones are only used for acquisition. In this work we depart from this setting by using both labeled and unlabeled data during model training across active learning cycles. We do so by using unsupervised feature learning at the beginning of the active learning pipeline and semi-supervised learning at every active learning cycle, on all available data. The former has not been investigated before in active learning, while the study of latter in the context of deep learning is scarce and recent findings are not conclusive with respect to its benefit. Our idea is orthogonal to acquisition strategies by using more data, much like ensemble methods use more models. By systematically evaluating on a number of popular acquisition strategies and datasets, we find that the use of unlabeled data during model training brings a spectacular accuracy improvement in image classification, compared to the differences between acquisition strategies. We thus explore smaller label budgets, even one label per class.

## 1 INTRODUCTION

*Active learning* (Settles, 2009) is an important pillar of machine learning but it has not been explored much in the context of *deep learning* until recently (Gal et al., 2017; Beluch et al., 2018; Wang et al., 2017; Geifman & El-Yaniv, 2017; Sener & Savarese, 2018). The standard active learning scenario focuses on training a model on few labeled examples alone, while unlabeled data are only used for acquisition, *i.e.*, performing inference and selecting a subset for annotation. This is the opposite of what would normally work well when learning a deep model from scratch, *i.e.*, training on a lot of data with some loss function that may need labels or not. At the same time, evidence is being accumulated that, when training powerful deep models, the difference in performance between acquisition strategies is small (Gissin & Shalev-Shwartz, 2018; Chitta et al., 2019; Beluch et al., 2018).

In this work, focusing on image classification, we revisit active deep learning with the seminal idea of using all data, whether labeled or not, during model training at each active learning cycle. This departs from the standard scenario in that unlabeled data are now directly contributing to the cost function being minimized and to subsequent parameter updates, rather than just being used to perform inference for acquisition, whereby parameters are fixed. We implement our idea using two principles: *unsupervised feature learning* and *semi-supervised learning*. While both are well recognized in deep learning in general, we argue that their value has been unexplored or underestimated in the context of deep active learning.

*Unsupervised feature learning* or *self-supervised learning* is a very active area of research in deep learning, often taking the form of pre-training on artificial tasks with no human supervision for representation learning, followed by supervised fine-tuning on different target tasks like classification or object detection (Doersch et al., 2015; Wang & Gupta, 2015; Gidaris et al., 2018; Caron et al., 2018). To our knowledge, all deep active learning research so far considers training deep models from scratch. In this work, we perform unsupervised feature learning on all data once at the beginning of the active learning pipeline and use the resulting parameters to initialize the model at each active learning cycle. Relying on Caron et al. (2018), we show that such unsupervised pre-training improves accuracy in many cases at little additional cost.

*Semi-supervised learning* (Chapelle et al., 2006) and active learning can be seen as two facets of the same problem: the former focuses on *most* certain model predictions on unlabeled examples, while the latter on *least* certain ones. Combined approaches appeared quite early (McCallum & Nigam, 1998; Zhu et al., 2003). In the context of deep learning however, such combinations are scarce (Wang et al., 2017) and have even been found harmful in cases (Ducoffe & Precioso, 2018). It has also been argued that the two individual approaches have similar performance, while active learning has lower cost (Gal et al., 2017). In the meantime, research on deep semi-supervised learning is very active, bringing significant progress (Tarvainen & Valpola, 2017; Laine & Aila, 2017; Iscen et al., 2019; Verma et al., 2019). In this work, we use semi-supervised learning on all data at every active learning cycle, replacing supervised learning on labeled examples alone. Relying on Iscen et al. (2019), and contrary to previous findings Wang et al. (2017); Gal et al. (2017), we show that this consistently brings a dramatic accuracy improvement.

Since Iscen et al. (2019) uses *label propagation* (Zhou et al., 2003a) to explore the manifold structure of the feature space, an important question is whether it is the manifold similarity or the use of unlabeled data during model training that actually helps. We address this question by introducing a new acquisition strategy that is based on label propagation.

In summary, we make the following contributions:

- We systematically benchmark a number of existing acquisition strategies, as well as a new one, on a number of datasets, evaluating the benefit of unsupervised pre-training and semi-supervised learning in all cases.
- Contrary to previous findings, we show that using unlabeled data during model training can yield a dramatic gain compared to differences between acquisition strategies.
- Armed with this finding, we explore a smaller budget (fewer labeled examples) than prior work, and we find that the random baseline may actually outperform all other acquisition strategies by a large margin in cases.

## 2    RELATED WORK

We focus on deep active and semi-supervised learning as well as their combination.

**Active learning.** *Geometric* methods like *core sets* (Geifman & El-Yaniv, 2017; Sener & Savarese, 2018) select examples based on distances in the feature space. The goal is to select a subset of examples that best approximate the whole unlabeled set. We introduce a similar approach where Euclidean distances are replaced by manifold ranking. There are methods inspired by *adversarial learning*. For instance, a binary classifier can be trained to discriminate whether an example belongs to the labeled or unlabeled set (Gissin & Shalev-Shwartz, 2018; Sinha et al., 2019). *Adversarial examples* have been used, being matched to the nearest unlabeled example (Mayer & Timofte, 2018) or added to the labeled pool (Ducoffe & Precioso, 2018).

It has been observed however that deep networks can perform similarly regardless of the acquisition function (Gissin & Shalev-Shwartz, 2018; Chitta et al., 2019), which we further investigate here. *Ensemble* and *Bayesian* methods (Gal et al., 2017; Beluch et al., 2018; Chitta et al., 2019) target representing model uncertainty, which than can be used by different acquisition functions. This idea is orthogonal to acquisition strategies. In fact, Beluch et al. (2018); Chitta et al. (2019) show that the gain of ensemble models is more pronounced than the gain of any acquisition strategy. Of course, ensemble and Bayesian methods are more expensive than single models. Approximations include for instance a single model producing different outputs by dropout (Gal et al., 2017). Our idea of using all data during model training is also orthogonal to acquisition strategies. It is also more expensive than using labeled data alone, but the gain is spectacular in this case. This allows the use of much smaller label budget for the same accuracy, which is the essence of active learning.

**Semi-supervised active learning** has a long history (McCallum & Nigam, 1998; Muslea et al., 2002; Zhu et al., 2003; Zhou et al., 2004; Long et al., 2008). A recent deep learning approach acquires the *least* certain unlabeled examples for labeling and at the same time assigns predicted *pseudo-labels* to *most* certain examples (Wang et al., 2017). This does not always help (Ducoffe & Precioso, 2018). In some cases, semi-supervised algorithms are incorporated as part of an active learning evaluation (Li et al., 2019; Sener & Savarese, 2018). A comparative study suggests that semi-supervised learning

does not significantly improve over active learning, despite its additional cost due to training on more data (Gal et al., 2017). We show that this is clearly not the case, using a state of the art semi-supervised method (Iscen et al., 2019) that is an inductive version of *label propagation* (Zhou et al., 2003a). This is related to Zhu & Ghahramani (2002); Zhu et al. (2003); Long et al. (2008), which however are limited to transductive learning.

**Unsupervised feature learning.** A number of unsupervised feature learning approaches pair matching images to learn the representation using a siamese architecture. These pairs can come as fragments of the same image (Doersch et al., 2015; Noroozi & Favaro, 2016) or as a result of tracking in video (Wang & Gupta, 2015). Alternatively, the network is trained on an artificial task like image rotation prediction (Gidaris et al., 2018) or even matching images to a noisy target (Bojanowski & Joulin, 2017). The latter is conceptually related to *deep clustering* (Caron et al., 2018), the approach we use in this work, where the network learns targets resulting from unsupervised clustering. It is interesting that in the context of semi-supervised learning, unsupervised pre-training has been recently investigated by Rebuffi et al. (2019), with results are consistent with ours. However, the use of unsupervised pre-training in deep active learning remains unexplored.

## 3 PROBLEM FORMULATION AND BACKGROUND

**Problem.** We are given a set $X := \{\mathbf{x}_i\}_{i \in \mathcal{I}} \subset \mathcal{X}$ of $n$ *examples* where $\mathcal{I} := [n] := \{1, \ldots, n\}$ and, initially, a collection $\mathbf{y}_0 := (y_i)_{i \in L_0}$ of $b$ *labels* $y_i \in C$ for $i \in L_0$, where $C := [c]$ is a set of $c$ classes and $L_0 \subset \mathcal{I}$ a set of indices with $|L_0| = b \ll n$. The goal of *active learning* (AL) (Settles, 2009) is to train a classifier in *cycles*, where in cycle $j = 0, 1, \ldots$ we use a collection $\mathbf{y}_j$ of labels for training, and then we *acquire* (or *sample*) a new batch $S_j$ of indices with $|S_j| = b$ to label the corresponding examples for the next cycle $j + 1$. Let $L_j := L_{j-1} \cup S_{j-1} \subset \mathcal{I}$ be the set of indices of *labeled* examples in cycle $j \geq 1$ and $U_j := \mathcal{I} \setminus L_j$ the indices of the *unlabeled* examples for $j \geq 0$. Then $\mathbf{y}_j := (y_i)_{i \in L_j}$ are the labels in cycle $j$ and $S_j \subset U_j$ is selected from the unlabeled examples. To keep notation simple, we will refer to a single cycle in the following, dropping subscripts $j$.

**Classifier learning.** The *classifier* $f_\theta : \mathcal{X} \to \mathbb{R}^c$ with parameters $\theta$, maps new examples to a vector of probabilities per class. Given $\mathbf{x} \in \mathcal{X}$, its *prediction* is the class of maximum probability

$$\pi(\mathbf{p}) := \arg\max_{k \in C} p_k, \tag{1}$$

where $p_k$ is the $k$-th element of vector $\mathbf{p} := f_\theta(\mathbf{x})$. As a by-product of learning parameters $\theta$, we have access to an *embedding function* $\phi_\theta : \mathcal{X} \to \mathbb{R}^d$, mapping an example $\mathbf{x} \in \mathcal{X}$ to a feature vector $\phi_\theta(\mathbf{x})$. For instance, $f_\theta$ may be a linear classifier on top of features obtained by $\phi_\theta$.

In a typical AL scenario, given a set of indices $L$ of labeled examples and labels $\mathbf{y}$, the *parameters* $\theta$ of the classifier are learned by minimizing the cost function

$$J(X, L, \mathbf{y}; \theta) := \sum_{i \in L} \ell(f_\theta(\mathbf{x}_i), y_i), \tag{2}$$

on labeled examples $\mathbf{x}_i$ for $i \in L$, where *cross-entropy* $\ell(\mathbf{p}, y) := -\log p_y$ for $\mathbf{p} \in \mathbb{R}_+^c$, $y \in C$.

**Acquisition.** Given the set of indices $U$ of unlabeled examples and the parameters $\theta$ resulting from training, one typically acquires a new batch by initializing $S \leftarrow \emptyset$ and then greedily updating by

$$S \leftarrow S \cup \{a(X, L \cup S, U \setminus S, \mathbf{y}; \theta)\} \tag{3}$$

until $|S| \geq b$. Here $a$ is an *acquisition* (or *sampling*) function, each time selecting one example from $U \setminus S$. For each $i \in S$, the corresponding example $\mathbf{x}_i$ is then given as query to an *oracle* (often a human expert), who returns a label $y_i$ to be used in the next cycle.

**Geometry.** Given parameters $\theta$, a simple acquisition strategy is to use the geometry of examples in the feature space $\mathcal{F}_\theta := \phi_\theta(\mathcal{X})$, without considering the classifier. Each example $\mathbf{x}_i$ is represented by the feature vector $\phi_\theta(\mathbf{x}_i)$ for $i \in \mathcal{I}$. One particular example is the function (Geifman & El-Yaniv, 2017; Sener & Savarese, 2018)

$$a(X, L, U, \mathbf{y}; \theta) := \arg\max_{i \in U} \min_{k \in L} \|\phi_\theta(\mathbf{x}_i), \phi_\theta(\mathbf{x}_k)\|, \tag{4}$$

each time selecting the unlabeled example in $U$ that is the most distant to its nearest labeled or previously acquired example in $L$. Such geometric approaches are inherently related to *clustering*. For instance, $k$-means++ (Arthur & Vassilvitskii, 2007) is a probabilistic version of (4).

**Uncertainty.** A common acquisition strategy that considers the classifier is some measure of uncertainty in its prediction. Given a vector of probabilities $\mathbf{p}$, one such measure is the *entropy*

$$H(\mathbf{p}) := -\sum_{k=1}^{c} p_k \log p_k, \tag{5}$$

taking values in $[0, \log c]$. Given parameters $\theta$, each example $\mathbf{x}_i$ is represented by the vector of probabilities $f_\theta(\mathbf{x}_i)$ for $i \in \mathcal{I}$. Then, acquisition is defined by

$$a(X, L, U, \mathbf{y}; \theta) := \arg\max_{i \in U} H(f_\theta(\mathbf{x}_i)), \tag{6}$$

effectively selecting the $b$ *most uncertain* unlabeled examples for labeling.

**Pseudo-labels.** It is possible to use more data than the labeled examples while learning. In Wang et al. (2017) for example, given indices $L$, $U$ of labeled and unlabeled examples respectively and parameters $\theta$, one represents example $\mathbf{x}_i$ by $\mathbf{p}_i := f_\theta(\mathbf{x}_i)$, selects the *most certain* unlabeled examples

$$\hat{L} := \{i \in U : H(\mathbf{p}_i) \le \epsilon\}, \tag{7}$$

and assigns pseudo-label $\hat{y}_i := \pi(\mathbf{p}_i)$ by (1) for $i \in \hat{L}$. The same cost function $J$ defined by (2) can now be used by augmenting $L$ to $L \cup \hat{L}$ and $\mathbf{y}$ to $(\mathbf{y}, \hat{\mathbf{y}})$, where $\hat{\mathbf{y}} := (\hat{y}_i)_{i \in \hat{L}}$. This augmentation occurs once per cycle in Wang et al. (2017). This is an example of active *semi-supervised* learning.

**Transductive label propagation** (Zhou et al., 2003a) refers to graph-based, semi-supervised learning. A nearest neighbor graph of the dataset $X$ is used, represented by a symmetric non-negative $n \times n$ *adjacency* matrix $W$ with zero diagonal. This matrix is symmetrically normalized as $\mathcal{W} := D^{-1/2} W D^{-1/2}$, where $D := \mathrm{diag}(W\mathbf{1})$ is the *degree* matrix and $\mathbf{1}$ is the all-ones vector. The given labels $\mathbf{y} := (y_i)_{i \in L}$ are represented by a $n \times c$ zero-one matrix $Y := \chi(L, \mathbf{y})$ where row $i$ is a $c$-vector that is a *one-hot* encoding of label $y_i$ if example $\mathbf{x}_i$ is labeled and zero otherwise,

$$\chi(L, \mathbf{y})_{ik} := \begin{cases} 1, & i \in L \land y_i = k, \\ 0, & \text{otherwise} \end{cases} \tag{8}$$

for $i \in \mathcal{I}$ and $k \in C$. Zhou et al. (2003a) define the $n \times c$ matrix $P := \eta[h(Y)]$[1], where

$$h(Y) := (1 - \alpha)(I - \alpha\mathcal{W})^{-1}Y, \tag{9}$$

$I$ is the $n \times n$ identity matrix, and $\alpha \in [0, 1)$ is a parameter. The $i$-th row $\mathbf{p}_i$ of $P$ represents a vector of class probabilities of unlabeled example $\mathbf{x}_i$, and a prediction can be made by $\pi(\mathbf{p}_i)$ (1) for $i \in U$. This method is *transductive* because it cannot make predictions on previously unseen data without access to the original data $X$.

**Inductive label propagation.** Although the previous methods do not apply to unseen data by themselves, the predictions made on $X$ can again be used as pseudo-labels to train a classifier. This is done in Iscen et al. (2019), applied to semi-supervised learning. Like Wang et al. (2017), a pseudo-label is generated for unlabeled example $\mathbf{x}_i$ as $\hat{y}_i := \pi(\mathbf{p}_i)$ by (1), only now $\mathbf{p}_i$ is the $i$-th row of the result $P$ of label propagation according to (9) rather than the classifier output $f_\theta(\mathbf{x}_i)$. Unlike Wang et al. (2017), *all* unlabeled examples are pseudo-labeled and an additional cost term $J_w(X, U, \hat{\mathbf{y}}; \theta) := \sum_{i \in U} w_i \ell(f_\theta(\mathbf{x}_i), \hat{y}_i)$ applies to those examples, where $\hat{\mathbf{y}} := (\hat{y}_i)_{i \in U}$ and $w_i := \beta(\mathbf{p}_i)$ is a weight reflecting the *certainty* in the prediction of $\hat{y}_i$:

$$\beta(\mathbf{p}) := 1 - \frac{H(\mathbf{p})}{\log c}. \tag{10}$$

Unlike Wang et al. (2017), the graph and the pseudo-labels are updated *once per epoch* during learning in Iscen et al. (2019), where there are no cycles.

---

[1] We denote by $\eta[A] := \mathrm{diag}(A\mathbf{1})^{-1} A$ and $\eta[\mathbf{a}] := (\mathbf{a}^\top \mathbf{1})^{-1}\mathbf{a}$ the (row-wise) $\ell_1$-normalization of nonnegative matrix $A$ and vector $\mathbf{a}$ respectively.

**Unsupervised feature learning.** Finally, it is possible to train an embedding function in an unsupervised fashion. A simple method that does not make any assumption on the nature or structure of the data is Caron et al. (2018). Simply put, starting by randomly initialized parameters $\theta$, the data $\phi_\theta(X)$ are *clustered* by $k$-means, each example is assigned to the nearest centroid, clusters and assignments are treated as classes $C$ and *pseudo-labels* $\hat{\mathbf{y}}$ respectively, and learning takes place according to $J(X, \mathcal{I}, \hat{\mathbf{y}}, \theta)$ (2). By updating the parameters $\theta$, $\phi_\theta(X)$ is updated too. The method therefore alternates between clustering/pseudo-labeling and feature learning, typically once per epoch.

## 4 TRAINING THE MODEL ON UNLABELED DATA

We argue that acquiring examples for labeling is not making the best use of unlabeled data: unlabeled data should be used during model training, appearing in the cost function that is being minimized. We choose two ways of doing so: unsupervised feature learning and semi-supervised learning. As outlined in Algorithm 1, we follow the standard active learning setup, adding unsupervised pre-training at the beginning and replacing supervised learning on $L$ by semi-supervised learning on $L \cup U$ at each cycle. The individual components are discussed in more detail below.

---

**Algorithm 1:** Semi-supervised active learning

**Data:** data $X$, indices of labeled examples $L$, labels $\mathbf{y}$, batch size $b$

1   $U \leftarrow \mathcal{I} \setminus L$          $\triangleright$ indices of unlabeled examples
2   $\theta_0 \leftarrow \text{PRE}(X)$          $\triangleright$ unsupervised pre-training
3   **for** $j \in \{0, \dots\}$ **do**          $\triangleright$ active learning cycles
4     $\theta \leftarrow \text{SUP}(X, L, \mathbf{y}; \theta_0)$          $\triangleright$ supervised learning on $L$ only
5     **for** $e \in \{1, \dots\}$ **do**          $\triangleright$ epochs
6       $(\hat{\mathbf{y}}, \mathbf{w}) \leftarrow \text{LP}(X, L, \mathbf{y}, \theta)$          $\triangleright$ pseudo-labels $\hat{\mathbf{y}}$ and labels $\mathbf{w}$
7       $\theta \leftarrow \text{SEMI}(X, L \cup U, (\mathbf{y}, \hat{\mathbf{y}}), \mathbf{w}; \theta)$    $\triangleright$ semi-supervised learning on all data
8     $S \leftarrow \emptyset$
9     **while** $|S| < b$ **do**          $\triangleright$ acquire a batch $S \subset U$ for labeling
10       $S \leftarrow S \cup a(X, L \cup S, U \setminus S, \mathbf{y}; \theta)$
11     $\mathbf{y} \leftarrow (\mathbf{y}, \text{LABEL}(S))$          $\triangleright$ obtain true labels on $S$ by oracle
12     $L \leftarrow L \cup S; U \leftarrow U \setminus S$          $\triangleright$ update indices

---

*Unsupervised pre-training* (PRE) takes place at the beginning of the algorithm. We follow Caron et al. (2018), randomly initializing $\theta$ and then alternating between clustering the features $\phi_\theta(X)$ by $k$-means and learning on cluster assignment pseudo-labels $\hat{\mathbf{y}}$ of $X$ according to $J(X, \mathcal{I}, \hat{\mathbf{y}}, \theta)$ (2). The result is a set of parameters $\theta_0$ used to initialize the classifier at every cycle.

Learning per cycle follows inductive label propagation (Iscen et al., 2019). This consists of supervised learning followed by alternating label propagation and semi-supervised learning on all examples $L \cup U$ at every epoch. The *supervised learning* (SUP) is performed on the labeled examples $L$ only using labels $\mathbf{y}$, according to $J(X, L, \mathbf{y}, \theta)$ (2), where the parameters $\theta$ are initialized by $\theta_0$.

*Label propagation* (LP) involves a reciprocal $k$-nearest neighbor graph on features $\phi_\theta(X)$ (Iscen et al., 2019). As in Zhou et al. (2003a), the resulting affinity matrix $W$ is normalized as $\mathcal{W} := D^{-1/2}WD^{-1/2}$. Label propagation is then performed according to $P = \eta[h(Y)]$ (9), by solving the corresponding linear system using the *conjugate gradient* (CG) method (Iscen et al., 2019). The label matrix $Y := \chi(L, \mathbf{y})$ (8) is defined on the true labeled examples $L$ that remain fixed over epochs but grow over cycles. With $\mathbf{p}_i$ being the $i$-th row of $P$, a pseudo-label $\hat{y}_i = \pi(\mathbf{p}_i)$ (1) and a weight $w_i = \beta(\mathbf{p}_i)$ (10) are defined for every $i \in U$ (Iscen et al., 2019).

*Semi-supervised learning* (SEMI) takes place on all examples $L \cup U = \mathcal{I}$, where examples in $L$ have true labels $\mathbf{y}$ and examples in $U$ pseudo-labels $\hat{\mathbf{y}} := (\hat{y}_i)_{i \in U}$. Different than Iscen et al. (2019), we minimize the standard cost function $J(X, L \cup U, (\mathbf{y}, \hat{\mathbf{y}}), \theta)$ (2), but we do take weights $\mathbf{w} := (w_i)_{i \in U}$ into account in mini-batch sampling, $\ell_1$-normalized as $\eta[\mathbf{w}]$. In particular, part of each mini-batch is drawn uniformly at random from $L$, while the other part is drawn with replacement from the discrete distribution $\eta[\mathbf{w}]$ on $U$: an example may be drawn more than once per epoch or never.

|  | Data size train / test | Image size | Mini-batch size w/o SEMI/ SEMI | Budget | Total labels |
|---|---|---|---|---|---|
| MNIST | 60000 / 10000 | $28 \times 28$ | 10 / 64 | 10 | 50 |
| SVHN | 73257 / 26032 | $32 \times 32$ | 32 / 128 | 100 | 500 |
| CIFAR-10 | 50000 / 10000 | $32 \times 32$ | 32 / 128 | 100 | 500 |
| CIFAR-10 | 50000 / 10000 | $32 \times 32$ | 32 / 128 | 1000 | 5000 |
| CIFAR-100 | 50000 / 10000 | $32 \times 32$ | 32 / 128 | 1000 | 5000 |

Table 1: Datasets used in this paper, including the mini-batch sizes used in training with and without SEMI, acquisition size at each active learning step and the total number of labeled images.

**Discussion.** The above *probabilistic weighting* decouples the size of the epoch from $n$ and indeed we experiment with epochs smaller than $n$, accelerating learning compared to Iscen et al. (2019). It is similar to *importance sampling*, which is typically based on loss values (Katharopoulos & Fleuret, 2017; Cheng et al., 2018) or predicted class probabilities (Yang et al., 2015). Acceleration is important as training on all examples is more expensive than just the labeled ones, and is repeated at every cycle. On the contrary, unsupervised pre-trained only occurs once at the beginning.

The particular choice of components is not important: any unsupervised representation learning could replace Caron et al. (2018) in line 2 and any semi-supervised learning could replace Iscen et al. (2019) in lines 4-7 of Algorithm 1. We keep the pipeline as simple as possible, facilitating comparisons with more effective choices in the future.

## 5 INVESTIGATING MANIFOLD SIMILARITY IN THE ACQUISITION FUNCTION

Label propagation (Zhou et al., 2003a; Iscen et al., 2019) is based on the manifold structure of the feature space, as captured by the normalized affinity matrix $\mathcal{W}$. Rather than just using this information for propagating labels to unlabeled examples, can we use it in the acquisition function as well? This is important in interpreting the effect of semi-supervised learning in Algorithm 1: is any gain due to the use of manifold similarity, or to training the model on more data?

*Joint label propagation* (jLP), introduced here, is an attempt to answer these questions. It is an acquisition function similar in nature to the geometric approach (4), with Euclidean distance replaced by manifold similarity. In particular, the $n$-vector $Y\mathbf{1}_c$, the row-wise sum of $Y = \chi(L, \mathbf{y})$ (8), can be expressed as $Y\mathbf{1}_c = \delta(L) \in \mathbb{R}^n$, where

$$\delta(L)_i := \left\{ \begin{array}{ll} 1, & i \in L, \\ 0, & \text{otherwise} \end{array} \right. \tag{11}$$

for $i \in \mathcal{I}$. Hence, in the terminology of *manifold ranking* (Zhou et al., 2003b), vector $Y\mathbf{1}_c$ represents a set of *queries*, one for each example $\mathbf{x}_i$ for $i \in L$, and the $i$-th element of the $n$-vector $h(Y)\mathbf{1}_c$ in (9) expresses the *manifold similarity* of $\mathbf{x}_i$ to the queries for $i \in \mathcal{I}$. Similar to (4), we acquire the example in $U$ that is the *least similar* to examples in $L$ that are labeled or previously acquired:

$$a(X, L, U, \mathbf{y}; \theta) := \arg\min_{i \in U}(h(\delta(L)))_i. \tag{12}$$

This strategy is only geometric and bears similarities to *discriminative active learning* (Gissin & Shalev-Shwartz, 2018), which learns a binary classifier to discriminate labeled from unlabeled examples and acquires examples of least confidence in the "labeled" class.

## 6 EXPERIMENTS

### 6.1 EXPERIMENTAL SETUP

**Datasets.** We conduct experiments on four datasets that are most often used in deep active learning: MNIST (LeCun et al., 1998), SVHN (Netzer et al., 2011), CIFAR-10 and CIFAR-100 (Krizhevsky, 2009). Table 1 presents statistics of the datasets. Following Tarvainen & Valpola (2017); Iscen et al. (2019), we augment input images by $4 \times 4$ random translations and random horizontal flips.

**Networks and training.** For all experiments we use a 13-layer convolutional network used previously in Laine & Aila (2016). We train the model from scratch at each active learning cycle, using

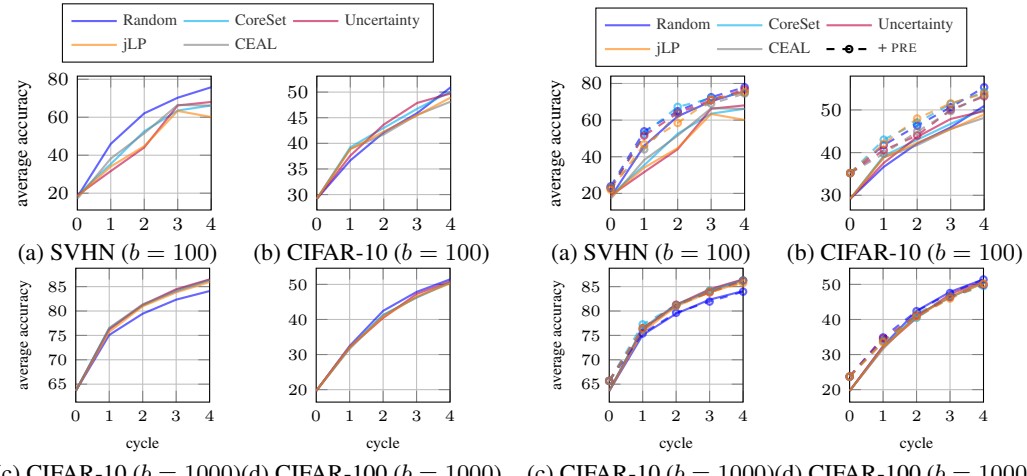

Figure 1: Average accuracy *vs.* cycle on different setups and acquisition strategies.

Figure 2: Average accuracy *vs.* cycle on different setups and acquisition strategies with PRE.

SGD with momentum of 0.9 for 200 epochs. An initial learning rate of 0.2 is decayed by cosine annealing (Loshchilov & Hutter, 2017), scheduled to reach zero at 210 epochs. The mini-batch size is 32 for standard training and 128 when SEMI is used, except for MNIST where the size of the mini-batch 10 and 64 with SEMI. All other parameters follow Tarvainen & Valpola (2017).

**Unsupervised pre-training.** We use $k$-means as the clustering algorithm and follow the settings of Caron et al. (2018). The model is trained for 250 epochs on the respective datasets.

**Semi-supervised learning.** Following Iscen et al. (2019), we construct a reciprocal $k$-nearest neighbor graph on features $\phi_\theta(X)$, with $k = 50$ neighbors and similarity function $s(\mathbf{u}, \mathbf{v}) := [\hat{\mathbf{u}}^\top \hat{\mathbf{v}}]^3_+$ for $\mathbf{u}, \mathbf{v} \in \mathbb{R}^d$, where $\hat{\mathbf{u}}$ is the $\ell_2$-normalized counterpart of $\mathbf{u}$, while $\alpha = 0.99$ in (9). We follow Iscen et al. (2019) in splitting mini-batches into two parts: 50 examples (10 for MNIST) are labeled and the remaining pseudo-labeled. For the latter, we draw examples using normalized weights as a discrete distribution. The epoch ends when $\frac{1}{2}|U|$ pseudo-labels have been drawn, that is the epoch is 50% compared to Iscen et al. (2019). Given that $|L| \ll |U|$ in most cases, the labeled examples are typically repeated more than once.

**Acquisition strategies.** We evaluate our new acquisition strategy jLP along with the following baselines: (a) *Random*; (b) *Uncertainty* based on entropy (5); (c) *CEAL* (Wang et al., 2017), combining entropy with pseudo-labels (7); (d) the greedy version of *CoreSet* (4) (Sener & Savarese, 2018; Geifman & El-Yaniv, 2017).

**Baselines.** For all acquisition strategies, we show results of the complete Algorithm 1 as well as the the *standard baseline*, that is without pre-training and only fully supervised on labeled examples $L$, and *unsupervised pre-training* (PRE) alone without semi-supervised. In some cases, we show *semi-supervised* (SEMI) alone. For instance, in the scenario of 100 labels per class, the effect of pre-training is small, especially in the presence of semi-supervised. CEAL (Wang et al., 2017) is a baseline with its own pseudo-labels, so we do not combine it with semi-supervised. The length of the epoch is fixed for Algorithm 1 and increases with each cycle.

**Label budget and cycles.** We consider three different scenarios, as shown in Table 1. In the first, we use an initial balanced label set $L_0$ of 10 labels per class, translating into a total of 100 for CIFAR-10 and SVHN and 1000 for CIFAR-100. We use the same values as label budget $b$ for all cycles. In the second, we use initially 100 labels per class in CIFAR-10 with $b = 1000$ per cycle; this is not interesting for CIFAR-100 as it results in complete labeling of the training set after 4 cycles. Finally, we investigate the use of one label per class both as the initial set and the label budget, on MNIST, translating to 10 labels per cycle. All experiments are carried out for 5 cycles and repeated 5 times using different initial label sets $L_0$. We report *average accuracy* and *standard deviation*.

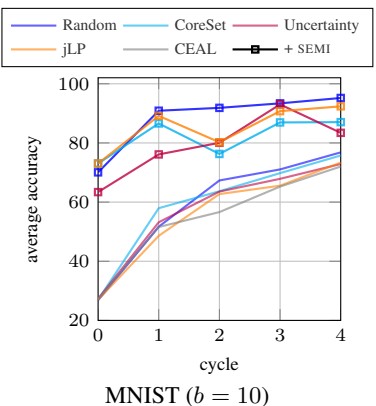

Figure 3: Average accuracy *vs.* cycle on different setups and acquisition strategies.

| METHOD | CIFAR-10 | CIFAR-100 |
|---|---|---|
| BUDGET | $b = 100$ | $b = 1000$ |
| CYCLE 0 | | |
| jLP | 29.17±1.62 | 19.63±0.99 |
| + PRE | 35.20±2.26 | 23.71±0.86 |
| + SEMI | 36.73±5.70 | 25.06±1.44 |
| + PRE + SEMI | **38.05±2.92** | **27.04±0.78** |
| CYCLE 1 | | |
| jLP | 38.86±1.36 | 32.16±1.98 |
| + PRE | 42.07±0.74 | 33.48±0.52 |
| + SEMI | 46.76±3.27 | 37.99±2.47 |
| + PRE + SEMI | **48.66±2.64** | **40.30±1.53** |
| CYCLE 2 | | |
| jLP | 42.30±1.61 | 40.65±1.21 |
| + PRE | 47.99±1.17 | 40.81±0.40 |
| + SEMI | **51.53±3.02** | 46.39±1.49 |
| + PRE + SEMI | 51.18±1.80 | **47.03±0.47** |

Table 2: Ablation study. Evaluation of results obtained with Random while adding PRE and/or SEMI.

## 6.2 STANDARD BASELINE RESULTS

We first evaluate acquisition functions without using any unlabeled data. Figure 1 presents results on SVHN, CIFAR-10 and CIFAR-100. The differences between acquisition functions are not significant, except when compared to Random. On SVHN, Random appears to be considerably better than the other acquisition functions and worse on CIFAR-10 with $b = 1000$. All the other acquisition functions give near identical results; in particular, there is no clear winner in the case of 10 labels per class on CIFAR-10 and CIFAR-100 (Figure 1(b) and (d), respectively).

This confirms similar observations made in Gissin & Shalev-Shwartz (2018) and Chitta et al. (2019). Our jLP is no exception, giving similar results to the other acquisition functions. We study this phenomenon in Appendix B. In summary, we find that while the ranks of examples according to different strategies may be uncorrelated, the resulting predictions of label propagation mostly agree. Even in cases of disagreement, the corresponding examples have small weights, hence their contribution to the cost function is small. Since those predictions are used as pseudo-labels in Iscen et al. (2019), this can explain why the performance of the learned model is also similar in the presence of semi-supervised learning.

## 6.3 THE EFFECT OF UNSUPERVISED PRE-TRAINING

As shown in Figure 2, pre-training can be beneficial. PRE by itself brings substantial gain on SVHN and CIFAR-10 with $b = 100$, up to 6%, while the improvements on CIFAR-100 are moderate. In addition, numerical results in Table 2 for our acquisition strategy jLP show that PRE is beneficial with or without SEMI in most cases. Pre-training provides a relatively easy and cost-effective improvement. It is performed only once at the beginning of the active learning process. While Caron et al. (2018) was originally tested on large datasets like ImageNet or YFCC100M, we show that it can be beneficial even on smaller datasets like CIFAR-10 or SVHN.

## 6.4 THE EFFECT OF SEMI-SUPERVISED LEARNING

Figure 4 shows results on different datasets and acquisition strategies like Figure 2, but including both PRE and PRE + SEMI. For the purpose of reproducibility, numeric results, including average and standard deviation measurements, are given in Appendix A for all cycles and datasets.

The combination PRE + SEMI yields a further significant improvement over PRE and the standard baseline, on all acquisition functions and datasets. For instance, on CIFAR-10 with a budget of 100, the most noticeable improvement comes from Random, where the improvement of PRE + SEMI is around 15% over the standard baseline at all cycles. The improvement is around 10% in most other cases, which is by far greater than any potential difference between the acquisition methods. Also,

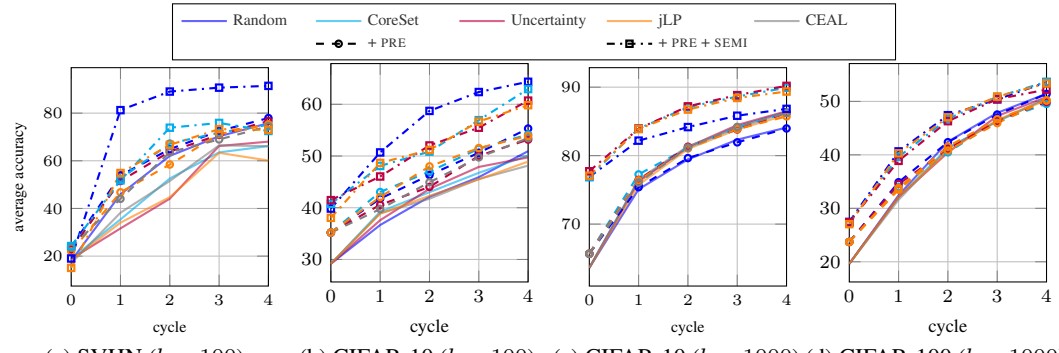

Figure 4: Average accuracy *vs.* cycle on different setups and acquisition strategies: Baseline, PRE and PRE +SEMI. PRE and PRE + SEMI scenarios are represented using different dashed lines as presented in the legend. For reference, the full training accuracy is 96.97% for SVHN, 94.84 % for CIFAR-10 and 76.43 % for CIFAR-100.

noticeably, in the case of SVHN, Random with PRE + SEMI reaches nearly the fully supervised accuracy after just 2 cycles (300 labeled examples in total).

The gain from semi-supervised learning is dramatic in the few-labels regime of CIFAR-10 with $b = 100$. A single cycle with PRE + SEMI achieves the accuracy of 4 cycles of the standard baseline in this case, which translates to a significant reduction of cost for human annotation.

In Table 2 we present the effect of all four combinations: with/without PRE and with/without SEMI. We focus on our jLP acquisition strategy, which has similar performance as all other strategies and uses manifold similarity just like SEMI. In most cases, PRE improves over SEMI alone by around 2%. The use of PRE appears to be particularly beneficial in the first cycles, while its impact decreases as the model performance improves.

It is worth noting that CEAL, which makes use of pseudo-labels, has a low performance. This has been observed before (Ducoffe & Precioso, 2018) and can be attributed to the fact that it is using the same set of pseudo-labels in every epoch. By contrast, pseudo-labels are updated in every epoch in our case.

## 6.5 LABEL PROPAGATION WITH ONE LABEL PER CLASS

Since PRE and SEMI have a significant gain in classification accuracy, it is reasonable to attempt even fewer labeled examples than in previous work on active learning. We investigate the extreme case of one label per class using MNIST as a benchmark, that is, label budget at each cycle is equal to the number of classes. Figure 3 shows results on all acquisition strategies with and without SEMI. As in the previous experiments, there is no consistent winner among the selection strategies alone, and accuracy remain below 80% after 5 cycles (50 labels in total) without SEMI. By contrast, Random with SEMI arrives at 90.89% accuracy after two cycles (20 labeled examples), which is 40% better than without SEMI.

## 7 DISCUSSION

In this work, we have shown the benefit of using both labeled and unlabeled data during model training in deep active learning for image classification. This leads to a more accurate model while requiring less labeled data, which is in itself one of the main objectives of active learning. We have used two particular choices for unsupervised feature learning and semi-supervised learning as components in our pipeline. There are several state of the art methods that could be used for the same purpose, for instance Tarvainen & Valpola (2017); Verma et al. (2019); Berthelot et al. (2019); Rebuffi et al. (2019) for semi-supervised learning. Our pipeline is as simple as possible, facilitating comparisons with more effective choices, which can only strengthen our results. While the improvement coming from recent acquisition strategies is marginal in many scenarios, an active learning approach that uses unlabeled data for training and not just acquisition appears to be a very

good option for deep network models. Our findings can have an impact on how deep active learning is evaluated in the future. For instance, the relative performance of the random baseline to all other acquisition strategies depends strongly on the label budget, the cycle and the presence of pre-training and semi-supervised learning.

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

## A    ALL DETAILED RESULTS

In order to facilitate reproducibility, in this section we present all the detailed results in Table 3 and Table 4. We describe results obtained with the five methods presented before, namely Random, Uncertainty, CEAL, CoreSet and jLP. We evaluate them on CIFAR-10 with 10 and 100 labels per class (budget $b = 100$ and $b = 1000$ respectively), CIFAR-100 with $b = 1000$ in Table 3. We present results obtained on MNIST with only 1 label per class ($b = 10$) and SVHN with $b = 100$ in Table 4.

## B    STUDYING THE AGREEMENT OF ACQUISITION STRATEGIES

It has been observed that most acquisition strategies do not provide a significant improvement over standard uncertainty when using deep neural networks; for instance, all strategies perform similarly on CIFAR-10 and CIFAR-100 according to Gissin & Shalev-Shwartz (2018) and Chitta et al. (2019). To better understand the differences, the ranks of examples acquired by different strategies are compared pairwise by Gissin & Shalev-Shwartz (2018). We make a step further in this direction, using label propagation as a tool.

### B.1    MEASURING AGREEMENT

After the classifier is trained at any cycle using any reference acquisition function $a$, we apply two different acquisition functions, say $a^{(1)}$ and $a^{(2)}$, followed by labeling of acquired examples and label propagation, obtaining two different sets of predicted pseudo-labels $\hat{\mathbf{y}}^{(1)}$ and $\hat{\mathbf{y}}^{(2)}$ and weights $\mathbf{w}^{(1)}$ and $\mathbf{w}^{(2)}$ on the unlabeled examples $U$. We define the *weighted accuracy*

$$A_{U,\mathbf{w}}(\mathbf{z}, \mathbf{z}') = \sum_{i \in U} \eta[\mathbf{w}]_i \delta_{z_i, z_i'} \tag{13}$$

for $\mathbf{z}, \mathbf{z}' \in \mathbb{R}^{|U|}$, where $\delta$ is the Kronecker delta function. Using the average weights $\mathbf{w} := \frac{1}{2}(\mathbf{w}^{(1)} + \mathbf{w}^{(2)})$, we then measure the weighted accuracy $A_{U,\mathbf{w}}(\mathbf{y}^{(1)}, \mathbf{y}^{(2)})$, expressing the *agreement* of the two strategies, as well as the weighted accuracy $A_{U,\mathbf{w}}(\mathbf{y}^{(k)}, \mathbf{t})$ of $a^{(k)}$ relative to the true labels $\mathbf{t}$ on $U$ for $k = 1, 2$. More measurements include weighted accuracies relative to true labels on subsets of $U$ where the two strategies agree or disagree. This way, assuming knowledge of the true labels on the entire set $X$, we evaluate the quality of pseudo-labels used in semi-supervised learning in each cycle, casting label propagation as an efficient surrogate of the learning process.

### B.2    RESULTS

We show results on CIFAR-10 with $b = 1000$ in this study. Following the experiments of Gissin & Shalev-Shwartz (2018), we first investigate the correlation of the ranks of unlabeled examples obtained by two acquisition functions. As shown in Figure 5(a), Uncertainty and jLP are not as heavily correlated compared to, for example, *CoreSet* and *Uncertainty* in Figure 5(b). The correlation between *jLP* and *CoreSet* is also quite low as shown in Figure 5(c).

It may of course be possible that two strategies with uncorrelated ranks still yield models of similar accuracy. To investigate this, we measure agreement as described above. Results are shown in Table 5. Uncertainty is used as a reference strategy, *i.e.* we train the model for a number of cycles using Uncertainty and then measure agreement and disagreement of another strategy to Uncertainty. After cycle 1, any two methods agree on around 80% of the pseudo-labels, while the remaining 20% have on average smaller weights compared to when the methods agree.

We reach the same conclusions from a similar experiment where we actually train the model rather than perform label propagation. Hence, although examples are ranked differently by different strategies, their effect on prediction, either by training or label propagation, is small.

| METHOD | CIFAR-10, $b = 100$ | | | CIFAR-10, $b = 1000$ | | CIFAR-100, $b = 1000$ | | |
|---|---|---|---|---|---|---|---|---|
| PRE | | ✓ | ✓ | | | | ✓ | ✓ |
| SEMI | | | ✓ | | ✓ | | | ✓ |
| CYCLE 0 | 100 LABELS | | | 1K LABELS | | 1K LABELS | | |
| Random | 29.17±1.62 | 35.20±2.26 | 39.84±2.63 | 63.61±1.42 | 78.85±0.86 | 19.63±0.99 | 23.71±0.86 | 27.46±0.52 |
| CYCLE 1 | 200 LABELS | | | 2K LABELS | | 2K LABELS | | |
| Random | 36.66±1.08 | 41.76±1.32 | **50.69±2.95** | 75.09±0.51 | 83.49±0.81 | **32.44±1.69** | **34.88±0.90** | 40.65±0.63 |
| Uncertainty | 37.59±1.93 | 40.56±2.21 | 46.04±2.78 | 76.22±0.68 | 84.94±0.35 | 32.09±1.50 | 34.54±0.70 | 38.88±1.11 |
| CoreSet | **39.23±1.17** | **43.04±0.92** | 48.08±1.64 | 76.44±0.34 | **84.98±0.19** | 32.05±1.40 | 33.95±0.57 | 39.63±0.70 |
| CEAL | 38.92±2.00 | 39.74±1.72 | – | **76.52±0.73** | – | 31.59±0.93 | 33.78±0.39 | – |
| jLP (ours) | 38.86±1.36 | 42.07±0.74 | 48.66±2.64 | 75.74±0.39 | 84.62±0.47 | 32.16±1.98 | 33.48±0.52 | 40.30±1.53 |
| CYCLE 2 | 300 LABELS | | | 3K LABELS | | 3K LABELS | | |
| Random | 42.12±1.83 | 46.31±1.40 | **58.72±4.04** | 79.45±0.56 | 85.33±0.42 | **42.45±0.90** | **42.37±0.53** | 47.42±0.53 |
| Uncertainty | **43.66±1.57** | 44.02±1.73 | 52.04±2.46 | **81.26±0.30** | **87.65±0.29** | 40.43±0.63 | 41.04±0.27 | 46.30±1.12 |
| CoreSet | 43.01±2.14 | 47.00±2.57 | 50.85±4.23 | 81.11±0.61 | 87.21±0.31 | 41.32±0.70 | 40.47±0.38 | 46.74±1.00 |
| CEAL | 41.74±1.15 | 44.92±2.09 | – | 81.37±0.54 | – | 41.19±0.41 | 41.55±0.45 | – |
| jLP (ours) | 42.30±1.61 | **47.99±1.17** | 51.18±1.80 | 80.97±0.40 | 87.16±0.44 | 40.65±1.21 | 40.81±0.40 | 47.03±0.47 |
| CYCLE 3 | 400 LABELS | | | 4K LABELS | | 4K LABELS | | |
| Random | 45.91±1.63 | 50.63±0.59 | **62.37±1.41** | 82.33±0.21 | 86.66±0.21 | **47.85±0.84** | **47.54±0.63** | 50.38±0.25 |
| Uncertainty | **47.89±1.78** | 50.03±1.38 | 55.47±2.10 | **84.47±0.49** | **89.32±0.24** | 47.26±0.79 | 46.39±0.81 | 50.42±0.24 |
| CoreSet | 46.75±2.41 | 51.40±1.99 | 56.93±2.90 | 84.27±0.36 | 88.75±0.45 | 46.22±0.39 | 46.34±0.92 | 50.85±0.32 |
| CEAL | 45.55±2.39 | 49.73±1.82 | – | 84.05±0.44 | – | 46.34±0.44 | 46.67±0.38 | – |
| jLP (ours) | 45.49±1.71 | **51.54±1.24** | 56.67±2.58 | 83.82±0.02 | 88.85±0.38 | 46.52±0.99 | 45.94±0.44 | **50.90±0.67** |
| CYCLE 4 | 500 LABELS | | | 5K LABELS | | 5K LABELS | | |
| Random | **50.94±1.75** | **55.31±1.28** | **64.35±1.37** | 84.10±0.10 | 87.23±0.21 | **51.43±0.56** | **51.40±0.47** | 53.58±0.64 |
| Uncertainty | 49.73±2.29 | 53.17±1.52 | 60.71±2.77 | **86.49±0.19** | **90.42±0.28** | 50.83±0.31 | 49.90±0.82 | 52.20±0.50 |
| CoreSet | 50.11±1.40 | 54.17±0.40 | 62.94±2.41 | 86.39±0.36 | 90.33±0.13 | 50.48±0.84 | 49.54±0.95 | **53.67±1.29** |
| CEAL | 48.14±1.24 | 53.46±1.27 | – | 86.31±0.23 | – | 50.62±0.28 | 50.18±0.60 | – |
| jLP (ours) | 48.93±2.22 | 53.89±1.42 | 59.83±4.02 | 85.94±0.38 | 89.91±0.28 | 50.24±0.93 | 50.20±0.44 | 53.37±0.64 |

Table 3: Average accuracy and standard deviation for different label budget $b$ and cycle on CIFAR-10 and CIFAR-100. Following Algorithm 1, we show the effect of unsupervised pre-training (PRE) and semi-supervised learning (SEMI) compared to the standard baseline.

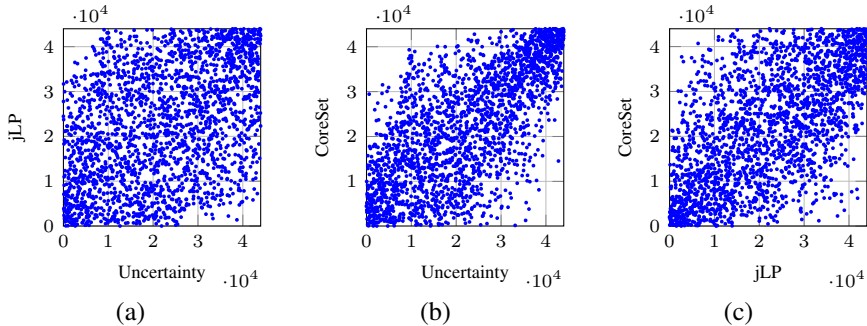

(a)    (b)    (c)

Figure 5: Ranks of examples obtained by one acquisition strategy *vs.* the ranks of another on CIFAR-10 with $b = 1000$ after cycle 1. A random 5% subset of all examples is shown.

| METHOD | MNIST, $b = 10$ | | | SVHN, $b = 100$ | |
|---|---|---|---|---|---|
| PRE | | | | ✓ | ✓ |
| SEMI | | ✓ | | | ✓ |
| CYCLE 0 | 10 LABELS | | | 100 LABELS | |
| Random | 26.83±4.15 | 70.06±12.87 | 18.00±2.47 | 23.83±4.63 | 19.01±5.61 |
| CYCLE 1 | 20 LABELS | | | 200 LABELS | |
| Random | 51.68±2.72 | **90.89±4.84** | 45.95±1.97 | **53.87±5.43** | **81.25±4.82** |
| Uncertainty | 53.18±5.88 | 76.12±11.07 | 31.63±8.75 | 51.52±2.36 | 37.84±21.00 |
| CoreSet | **57.94±7.16** | 86.59±10.98 | 35.39±7.16 | 52.49±5.76 | 51.80±10.62 |
| CEAL | 51.57±3.18 | – | 38.21±2.70 | 44.04±4.56 | – |
| jLP (ours) | 48.60±3.15 | 89.16±5.53 | 34.04±4.75 | 46.78±5.18 | 54.88±22.90 |
| CYCLE 2 | 30 LABELS | | | 300 LABELS | |
| Random | **67.31±5.19** | **91.86±3.89** | **62.05±3.23** | 64.88±4.93 | **89.05±2.07** |
| Uncertainty | 63.55±2.67 | 80.05±13.29 | 44.09±13.49 | 63.85±3.55 | 64.14±6.36 |
| CoreSet | 63.66±3.84 | 76.28±15.38 | 52.59±9.20 | **67.23±3.01** | 73.88±13.94 |
| CEAL | 56.62±7.05 | – | 51.53±5.93 | 63.58±2.80 | – |
| jLP (ours) | 62.71±2.82 | 80.23±4.11 | 44.74±17.50 | 58.43±9.82 | 66.68±13.91 |
| CYCLE 3 | 40 LABELS | | | 400 LABELS | |
| Random | **71.05±1.66** | **93.38±3.99** | **70.28±1.67** | **72.50±2.05** | **90.69±0.73** |
| Uncertainty | 67.87±3.26 | 93.03±4.88 | 66.21±3.68 | 70.90±2.48 | 56.60±5.69 |
| CoreSet | 69.79±3.36 | 86.93±7.62 | 63.53±6.34 | 71.79±3.58 | 75.88±6.95 |
| CEAL | 65.24±7.43 | – | 66.48±2.80 | 68.95±2.06 | – |
| jLP (ours) | 65.55±4.01 | 90.75±5.76 | 63.33±9.59 | 71.20±2.93 | 73.28±11.69 |
| CYCLE 4 | 50 LABELS | | | 500 LABELS | |
| Random | **76.81±2.19** | **95.20±3.61** | **75.78±1.90** | **77.93±1.55** | **91.44±0.80** |
| Uncertainty | 72.88±5.82 | 83.42±5.93 | 68.04±6.58 | 76.70±1.11 | 55.42±10.49 |
| CoreSet | 75.76±3.93 | 87.04±6.44 | 66.17±16.11 | 75.11±3.40 | 72.51±9.99 |
| CEAL | 72.02±7.96 | – | 66.14±14.42 | 74.48±1.98 | – |
| jLP (ours) | 73.36±4.43 | 92.37±5.38 | 60.12±20.06 | 75.33±1.44 | 72.98±12.01 |

Table 4: Average accuracy and standard deviation for different label budget $b$ and cycle on MNIST and SVHN. Following Algorithm 1, we show the effect of unsupervised pre-training (PRE) and semi-supervised learning (SEMI) compared to the standard baseline.

| CYCLE | | 1 | | | | 2 | | | |
|---|---|---|---|---|---|---|---|---|---|
| MEASURE | %agree | accuracy (13) | | avg weights | | %agree | accuracy (13) | | avg weights | |
| AGREE? | | = | ≠ | = | ≠ | | = | ≠ | = | ≠ |
| Random | 79.98 | 79.97 | 38.39 | 0.32 | 0.17 | 86.98 | 88.07 | 39.77 | 0.46 | 0.28 |
| CoreSet | 80.58 | 79.52 | 44.57 | 0.27 | 0.16 | 87.32 | 87.94 | 43.80 | 0.45 | 0.29 |
| jLP (ours) | 80.24 | 80.03 | 48.79 | 0.27 | 0.15 | 86.96 | 88.12 | 45.55 | 0.43 | 0.27 |

Table 5: Agreement results between acquisition strategies on CIFAR-10 with $b = 1000$ after cycles 1 and 2. All strategies are compared to Uncertainty as reference, which is also employed in the previous cycles. $\%agree$ is percentage of pseudo-labels agreeing to the reference. Accuracy is weighted according to (13) and weights are according to (10). Measurements denoted by $=$ ($\neq$) refer to the set of pseudo-labels that agree (disagree) with the reference.

