# OpenReview forum: "Rethinking deep active learning: Using unlabeled data at model training"
_ICLR.cc/2020/Conference — Reject_

### Official Review · AnonReviewer3 · 2019-10-19
**Official Blind Review #3**

**Rating:** 3

**Review:**

This paper argues that active learning (AL) methods shold combine unsupervised and semi-supervised learning during the iterative training process. Combining these complementary is indeed sensible, and this work is therefore a welcome effort. However, the results are quite mixed, and in fact seem to suggest that AL is rather ineffective. Therefore, what one might take from these results is that unsupervised and semi-supervised learning methods can boost predictive performance; but I think this is widely appreciated already. Perhaps a better framing for this work is: AL using standard metrics seems to be comparatively ineffective, especially when one uses pre-training/semi-supervised learning.

Some specific comments and questions:

- The authors have decided to frame this paper in terms of improving AL using un/semi-supervised learning. But given that, by the authors' own admission, the "random baseline may actually outperform all other acquisition strategies by a large margin", what is the motivation for adopting "AL" at all? I mean, if we are performing random (iid) sampling, this just reduces to vanilla learning with pre-training and semi-supervision; the 'active' component becomes irrelevant.

- I think the characterization of AL is not quite right on page 2. The authors write that AL is focuses on the "least certain" instances. This is often true -- namely under the popular uncertainty sampling regime -- but not all acquisition strategies use this heuristic. Indeed, even the geometry method the authors use explicitly ignores classifier confidence.

- The use of sampling in the SSL component is interesting, although an ablation here investigating this specific choice (as opposed to, say, naive sampling with uniform probability over unlabeled instances).

- I would not characterize the gains brought by unlabeled data here as "spectacular".

- As is often the case in work on AL, there is no real notion of a 'test set' here; instead the authors repeat experiments using different seed label sets. It is not entirely clear how much hyperparameter/architecture fine tuning was performed informally, but there is a lot going on here, so I would assume at least some. Therefore there is a risk that all results reported are in some sense optimistic, potentially being "overfit" to these datasets. It would be best to provide additional comparisons of approaches on completely unseen datasets.

**Experience Assessment:**

I have published in this field for several years.

**Review Assessment: Checking Correctness Of Derivations And Theory:**

I assessed the sensibility of the derivations and theory.

**Review Assessment: Checking Correctness Of Experiments:**

I assessed the sensibility of the experiments.

**Review Assessment: Thoroughness In Paper Reading:**

I read the paper at least twice and used my best judgement in assessing the paper.

---

> ### Author Response · Authors · 2019-11-08
> **Response to Reviewer #3 (Part 1/2)**
>
> Thank you for your review! Please find the response below.
>
> 1. "one might take from these results is that unsupervised and semi-supervised learning methods can boost predictive performance; but I think this is widely appreciated already."
>
> This is not exactly our main message. Of course unsupervised and semi-supervised learning methods are effective by themselves, and there is a lot of recent progress as discussed in our related work. The main point is the effect of unsupervised and semi-supervised learning methods when used in AL, relative to the effect of acquisition strategies (which are the core of AL) and the differences thereof. This is not appreciated as much. On the contrary, we clearly discuss cases like [Wang et al., 2017; Ducoffe & Precioso, 2018; Gal et al., 2017] where the combination of semi-supervised and AL is found not so effective or even harmful.
>
> 2. "Perhaps a better framing for this work is: AL using standard metrics seems to be comparatively ineffective, especially when one uses pre-training/semi-supervised learning."
>
> Exactly. This is our main message. In the abstract for instance, we say 'we find that the use of unlabeled data during model training brings a spectacular accuracy improvement in image classification, compared to the differences between acquisition strategies.' (See below about the criticism on "spectacular".)
>
> We are not the first to observe the similar performance of different acquisition strategies in the context of deep learning. See for instance [Gissin & Shalev-Shwartz, 2018; Chitta et al., 2019; Beluch et al.,2018], discussed extensively in our paper. We confirm their findings and in addition, as a main contribution, we systematically evaluate a number of strategies in the presence or not of unsupervised and semi-supervised learning, showing the relative effectiveness of all ideas in the same experimental setup.
>
> 3. "given that, by the authors' own admission, the "random baseline may actually outperform all other acquisition strategies by a large margin", what is the motivation for adopting "AL" at all? I mean, if we are performing random (iid) sampling, this just reduces to vanilla learning with pre-training and semi-supervision; the 'active' component becomes irrelevant."
>
> Exactly. 'Random baseline' means, as always, 'no AL'. So, overall, we evaluate the learning process with/without all three components: unsupervised, semi-supervised, active. In addition, we consider several acquisition strategies in the case of active.
>
> Now, on the actual result: indeed, when the label budget is small, random may be better than any other strategy. For instance, Fig. 1(a), SVHN (b=100). This we attribute to the weak representation obtained from few labels. The effect is amplified in the +PRE+SEMI version, Fig. 4(a). In the case of CIFAR-10 (b=100), random and other strategies are all similar in Fig. 1(b), but in the +PRE+SEMI version, Fig. 4(b), this effect happens again. Now, by comparing Fig 4(b) with 4(c), one realizes that random and the other strategies actually cross at some point around 1000 labels, after which AL becomes effective. This is a non-trivial result that can be very helpful in improving AL strategies.
>
> Therefore, our message is not necessarily negative: our recommendation is that unsupervised and semi-supervised should be used in the evaluation of AL methods and this may give rise to ideas for improved strategies, especially in the few label regime. We believe it is beyond the scope of this work to investigate such ideas.

---

> ### Author Response · Authors · 2019-11-08
> **Response to Reviewer #3 (Part 2/2)**
>
> 4."The use of sampling in the SSL component is interesting, although an ablation here investigating this specific choice."
>
> On CIFAR-10 (b=1000) +SEMI, cycle 1, uniform sampling without weights gives 77.6% accuracy, uniform with weights gives 78.4% and our approach gives 78.9% on average over 5 runs. This is a small improvement. We are adding more results.
>
> 5."I think the characterization of AL is not quite right on page 2. The authors write that AL is focuses on the "least certain""
>
> Of course. This could be e.g. "least certain" (considering the classifier) or "furthest" (considering the geometry of feature space alone). Whatever the criterion, AL and SSL can still be seen as two facets of the same problem. "Least certain" was an example. We are rephrasing.
>
> 6. "Spectacular".
>
> Well, in Fig. 4(a), cycle 0 for instance, Random+PRE+SEMI is better than Uncertainty alone by more than 50% (81.25 vs. 31.63 in Table 4). In Fig. 4(b), cycle 2, this difference is 15% (58.72 vs. 43.66 in Table 4). Differences between different acquisition strategies are rarely above 2%. We do find this spectacular, but we are rephrasing nevertheless.
>
> 7. "As is often the case in work on AL, there is no real notion of a 'test set' here; instead the authors repeat experiments using different seed label sets."
>
> We do not understand this comment. Could you please elaborate? Is maybe 'validation set' really meant?
>
> 8. "It is not entirely clear how much hyperparameter/architecture fine tuning was performed informally, but there is a lot going on here, so I would assume at least some. Therefore there is a risk that all results reported are in some sense optimistic, potentially being "overfit" to these datasets. It would be best to provide additional comparisons of approaches on completely unseen datasets."
>
> We assume a realistic scenario where only a handful of initial labels is given and therefore we opt to not use a validation set. To that end, we use the parameters specified in prior work like [Iscen et al., 2019], making changes only based on constraints of the protocol. For instance,
> - we increase the learning rate for faster convergence in all cases, even if this may be suboptimal;
> - we adjust the mini-batch size such that an epoch consists of several mini-batches in the 100 scenario;
> - we further reduce the mini-batch size in the extreme case of MNIST (b=10) scenario.
> Most importantly, hyperparameters are the same across all datasets and all scenarios. Overall, the argument of completely unseen datasets appears to apply to any work on AL, not just ours.

---

### Official Review · AnonReviewer2 · 2019-10-22
**Official Blind Review #2**

**Rating:** 3

**Review:**

This paper explores the setting where unsupervised/semi-supervised learning is combined with active learning. The results are that active learning doesn't really help. This paper is interesting in that it provides additional experiments for the intersection of active learning and unsupervised/semi-supervised learning. However, I don't really see the point of this paper. Active learning and unsupervised/semi-supervised learning have been combined before and there are other papers submitted to ICLR this year that combine these. The paper does not claim to provide anything new algorithmically (other than jLP which appears to work no better than random and isn't really advertised as the point of this paper). The only conclusion that I can draw is that sometimes unsupervised/semi-supervised learning works better than active learning, but no understanding of when and why this is the case (from other papers, it is not always the case).


Comments:

 - Although the paper claims to yield a general framework, it only does so partially. For instance, the framework in this work is restricted to semi-supervised methods that use pseudo-labels.

 - It may be the case that active learning doesn't help or even hurts because the batch size is too large and/or the initial seed set size is too small. Although this paper varies the acquisition strategies, these other hyper-parameters are equally, if not more, important.

**Experience Assessment:**

I have published one or two papers in this area.

**Review Assessment: Checking Correctness Of Derivations And Theory:**

N/A

**Review Assessment: Checking Correctness Of Experiments:**

I assessed the sensibility of the experiments.

**Review Assessment: Thoroughness In Paper Reading:**

I read the paper at least twice and used my best judgement in assessing the paper.

---

> ### Author Response · Authors · 2019-11-08
> **Response to Reviewer #2**
>
> Thank you for your review! Please find the response below.
>
> 1. "The only conclusion that I can draw is that sometimes unsupervised/semi-supervised learning works better than active learning, but no understanding of when and why this is the case (from other papers, it is not always the case)."
>
> Unsupervised/semi-supervised learning helps in all cases and the gain is significantly greater than the differences between AL acquisition strategies. Moreover, in certain cases of few labels (small budget b), all acquisition strategies are outperformed by Random, an effect that is amplified by the presence of unsupervised/semi-supervised learning.
>
> We cannot see where the conclusion above is drawn from. Could the reviewer please elaborate? What other papers are meant?
>
> 2. "the framework in this work is restricted to semi-supervised methods that use pseudo-labels."
>
> In each cycle, a new set of labels becomes available. The set of labeled example grows and there is also a set of unlabeled examples. At this point, a model is learned. Standard AL uses only the labeled examples to learn the model. Semi-supervised methods use the unlabeled examples too. There is no constraint as to what semi-supervised method one can choose. We chose a method that uses pseudo-labels. Any other method could be used.
>
> 3. "It may be the case that active learning doesn't help or even hurts because the batch size is too large and/or the initial seed set size is too small."
>
> We investigated different label budget scenarios including choices commonly used in AL papers (budget of 1k) as well as providing extreme cases (budget of 10). In general, we actually use equal or smaller budgets than prior work, because few labels is the most interesting case. The initial seed set is the same as the label budget per cycle according to the standard protocol. According to [Gissin & Shalev-Shwartz; 2018], differences between acquisition functions are even smaller when using a larger budget (5k).
>
> 4. "Active learning and unsupervised/semi-supervised learning have been combined before ..."
>
> Please see R1 point 6.
>
> 5. "... there are other papers submitted to ICLR this year that combine these."
>
> Maybe the reviewer wants to reconsider this comment.

---

### Official Review · AnonReviewer1 · 2019-11-03
**Official Blind Review #1**

**Rating:** 3

**Review:**

The authors study the problem of incorporating unsupervised (representation pre-training) learning and semi-supervised learning into active learning for image classification; specifically, performing pre-training before active learning starts [Caron, et al., 2018] and then applying inductive label propagation [Issen, et al., 2019] (slightly modification in the cost function to look more like importance sampling) before active learning querying occurs for each round (Algorithm 1).  The most novel technical innovation of this submission is the joint label propagation (jLP) querying function (which is a method of ‘spanning’ the learned manifold space).  Experiments are conducted on four (multi-class) image classification datasets (MNIST, SVHN, CIFAR-10, CIFAR-100), showing that unsupervised learning and semi-supervised learning can improve active learning on these datasets — although random selection often works better (as best as I can tell) implying that negative results are also a contribution of this paper. Finally, some active learning experiments are conducted using a per-round label budget of one example per class — also demonstrating mixed results with random sampling performing better in general.

In my mind, this paper has two primary components: (1) taking the position that semi-supervised and unsupervised learning can improve overall performance and, in principle, help with active learning and (2) propose jLP, which is a learning algorithm agnostic approach to spanning the manifold space. However, jLP doesn’t really seem to work in general. Thus, the main result is the first point — updating previous (pre-deep learning) results on SS/US AL to deep learning. Honestly, I think the primary conclusion is that semi-supervised and unsupervised learning has improved over the past decade (especially semi-supervised learning for image classification). The second result is that active learning in deep learning (at least for this application) hasn’t kept up. Wrt to (1), as the authors have pointed out, many others have applied semi-supervised learning to AL (including more that the authors didn’t include). Additionally, many have used unsupervised learning for AL (which the authors seem less aware of) from pre-clustering (e.g., [Nguyen & Smeulders, Active Learning using Pre-clustering; ICML04]) to one/few-shot learning (e.g., [Woodward & Finn, Active One-Shot Learning; NeurIPS16 workshop]) to using pre-trained embeddings for many ‘real-world tasks’ (e.g., NER [Shen, et al., Deep Active Learning for Named Entity Recognition; ICLR18] using word2vec). Thus, the interesting question would be to compare multiple pre-training techniques and ideally the relative effect on the active learning component (assuming this is the focus of the paper). With respect to semi-supervised learning, they have validated that inductive label propagation [Issen, et al., 2019] works for this task, but haven’t shown that this helps with active learning. Since this is a negative results without a theoretical contribution, I would again expect trying several semi-supervised algorithm and evaluating their relative performance in general and wrt the active learning querying strategy. Accordingly, I don’t think the contribution of this work in its current state is sufficiently well-developed — and would lean toward rejecting in its current form.

Below are some additional detailed comments (some also covered above):
— Given that this points toward a negative result, a more convincing direction to take would be to consider more combinations of unsupervised and semi-supervised approaches — specifically emphasizing how they affect the active learning component. This might point to more general findings and maybe toward a theory (maybe even consider a second application).
— The empirical emphasis is more around overall performance rather than the interaction between unsupervised representation learning and active learning, which is more toward the stated goal of the paper.
— Wouldn’t the right way to do (deep) representation learning in multiple rounds be to fine-tune at least some fraction of the time?  If the only claim is pre-training or pre-clustering, people certainly do this — just often not as a point of emphasis.
— The ‘first semi-supervised’ claim really only holds in the context of deep learning; however, scope is really more like semi-supervised applied to image classification, which would be a pretty narrow contribution in scope.
— Overall, there is a general overstatement of contributions and results: this is certainly not the first SSAL or USAL and the statement relative to deep learning is subtle; some of the empirical results are interesting, but I am not sure about ‘spectacular gains’ (and these gains aren’t seemingly due to the contribution of the paper).
— I don’t understand the ensemble model analogy in the abstract; is it because it is a ‘meta-algorithm’?

Some more positive notes:
+ It is interesting that there is some contradictory evidence relative to [Wang, et al., 2017; Gal, et al., 2017]; this is probably worth digging into a bit deeper.
+ The experimental details well-described given space constraints.

In summary, there are some interesting observations that are probably worth pursuing. However, the current contribution is basically that: (1) active learning doesn’t seem to really help, (2) semi-supervised learning and unsupervised learning improve performance for this task. Since (1) was really the point of the paper (as stated) in the title, I don’t think there is enough here to accept in its current form.

**Experience Assessment:**

I have published in this field for several years.

**Review Assessment: Checking Correctness Of Derivations And Theory:**

N/A

**Review Assessment: Checking Correctness Of Experiments:**

I carefully checked the experiments.

**Review Assessment: Thoroughness In Paper Reading:**

I read the paper thoroughly.

---

> ### Author Response · Authors · 2019-11-08
> **Response to Reviewer #1 (Part 1/2)**
>
> Thank you for your review. Please find our response below.
>
> 1. "... the interesting question would be to compare multiple pre-training techniques and ideally the relative effect on the active learning component (assuming this is the focus of the paper). ... Since this is a negative results without a theoretical contribution, I would again expect trying several semi-supervised algorithm and evaluating their relative performance in general and wrt the active learning querying strategy. Accordingly, I don’t think the contribution of this work in its current state is sufficiently well-developed - and would lean toward rejecting in its current form."
>
> The focus is indeed AL, this is why we consider several options for acquisition function. The focus is not unsupervised or semi-supervised learning, this is why we make a single choice for each. As we explicitly discuss in section 7, 'Our pipeline is as simple as possible, facilitating comparisons with more effective choices, which can only strengthen our results'. Stated otherwise, there is at least one choice of unsupervised and semi-supervised learning that yields significantly greater gain than any AL acquisition function over Random, or even makes all AL acquisition functions significantly inferior to Random in certain cases in the few label regime. This raises the question of rethinking at least how we should evaluate deep AL methods, as implied by the title. Any stronger unsupervised pre-training or semi-supervised method could only increase the gain, thus strengthening our conclusions.
>
> Considering that we already experiment on several acquisition functions and cycles, several datasets and label budgets, with/without PRE, with/without SEMI, adding any more options would also make our experiments cluttered; the plots of Fig. 4 are already hardly readable.
>
> We cannot follow the argument that a negative empirical result needs to be compensated by exhaustive sets of experiments. If a future work needs to validate that a new acquisition function outperforms others even in the presence of unsupervised/semi-supervised learning, would that validation need to include several options too?
>
> 2. "However, the current contribution is basically that: (1) active learning doesn’t seem to really help, (2) semi-supervised learning and unsupervised learning improve performance for this task. Since (1) was really the point of the paper (as stated) in the title, I don’t think there is enough here to accept in its current form."
>
> Does that mean that negative results are not welcome?
>
> 3. "With respect to semi-supervised learning, they have validated that inductive label propagation [Issen, et al., 2019] works for this task, but haven’t shown that this helps with active learning. ... The empirical emphasis is more around overall performance rather than the interaction between unsupervised representation learning and active learning, which is more toward the stated goal of the paper."
>
> It is known that unsupervised pre-training and semi-supervised learning help. The same is known for AL. What is not known is what is the relative gain of each of the three components on the same experimental setup. Algorithm 1 is exactly a combination of the three components and different combinations are systematically evaluated across all datasets, label budgets, and acquisition strategies. Our work is exactly on the interaction of the different components:
> - The finding that AL strategies are all outperformed by Random in certain cases of limited labels, indicates the effect of the quality of the representation on AL.
> - The a new acquisition function (jLP), more than a technical innovation, is exactly investigating whether manifold similarity (an idea coming from label propagation) helps in acquisition (which it doesn't, in line with our claim that unlabeled data should contribute to parameter updates).
> - The study of Appendix B investigates the effect of selected examples on label propagation, partially explaining why different acquisition strategies perform similarly, at least in the presence of label propagation.

---

> ### Author Response · Authors · 2019-11-08
> **Response to Reviewer #1 (Part 2/2)**
>
> 4. "many have used unsupervised learning for AL (which the authors seem less aware of) from pre-clustering (e.g., [Nguyen & Smeulders, Active Learning using Pre-clustering; ICML04]) to one/few-shot learning (e.g., [Woodward & Finn, Active One-Shot Learning; NeurIPS16 workshop]) to using pre-trained embeddings for many ‘real-world tasks’ (e.g., NER [Shen, et al., Deep Active Learning for Named Entity Recognition; ICLR18] using word2vec)."
>
> [Nguyen & Smeulders; ICML04], by propagating predictions within clusters, is more related to semi-supervised than unsupervised representation learning. This is a 2004 paper using a linear SVM classifier on raw images. We cannot see how [Woodward & Finn; NeurIPS16 workshop] is related to unsupervised pre-training. [Shen, et al.; ICLR18] is indeed relevant in that word2vec embeddings are used to initialize parameters that are subsequently updated. However, it is not the focus of this work to evaluate how this initialization helps compared to random initialization. It is very interesting to note how acquisition functions again perform similarly in this very different task as shown in Fig. 4. We shall discuss.
>
> 5. "The ‘first semi-supervised’ claim really only holds in the context of deep learning; however, scope is really more like semi-supervised applied to image classification, which would be a pretty narrow contribution in scope."
>
> Do we claim anything about ‘first semi-supervised’?
>
> We do make it clear that we consider the context of deep learning and that we focus on image classification. We do not find this scope narrow given the quantity of related work on AL focusing on the same task.
>
> 6. "Overall, there is a general overstatement of contributions and results: this is certainly not the first SSAL or USAL and the statement relative to deep learning is subtle; some of the empirical results are interesting, but I am not sure about ‘spectacular gains’ (and these gains aren’t seemingly due to the contribution of the paper)."
>
> SSAL is definitely not a claimed contribution. On USAL, see point 4 above. The gains are definitely not due to our contribution. However, as discussed, recent work has failed to take advantage of semi-supervised methods in AL [Gissin & Shalev-Shwartz, 2018; Chitta et al., 2019; Beluch et al.,2018]. CEAL for instance shows no improvement, while our choices yield significant gains (10-20% compared to 2-3% of differences between AL methods). See also R3 point 6 on "spectacular". As stated, our primary contribution is to 'systematically benchmark a number of existing acquisition strategies, ... on a number of datasets, evaluating the benefit of unsupervised pre-training and semi-supervised learning in all cases'.
>
> 7. "Wouldn’t the right way to do (deep) representation learning in multiple rounds be to fine-tune at least some fraction of the time?  If the only claim is pre-training or pre-clustering, people certainly do this — just often not as a point of emphasis."
>
> When training on the entire training set of labeled and unlabeled examples, fine-tuning seems indeed reasonable. However, we follow a standard protocol [Chitta et al., 2019] which involves training from scratch in each cycle. This keeps our pipeline simple and makes our work comparable to all previous work. It is also argued that training from scratch helps to avoid local minima potentially reached by the previous model due to the smaller number of labels. In fact, we have experimented with fine-tuning and our preliminary findings were that indeed performance was slightly inferior, while gains in speed were not significant as convergence was slow. We did not pursue this further as it was not the main focus of this work.
>
> 8. "I don’t understand the ensemble model analogy in the abstract; is it because it is a ‘meta-algorithm’?"
>
> Using ensemble models is an idea orthogonal to the choice of AL acquisition functions and improves performance in all cases, much like integrating unsupervised and semi-supervised methods into the AL pipeline.

---

### Public Comment · ~Thomas_Brox1 · 2019-10-20
**Indeed, some serious rethinking is necessary about deep active learning**

The paper title is a very friendly description of the devastating results (for active learning). We recently had the same finding with a slightly different experimental design: semi-supervised learning consistently outperforms active learning on the same data pool by a large margin, and the combination of semi-supervised and active learning is hardly better than just semi-supervised learning alone.

---

> ### Author Response · Authors · 2019-10-28
> **Thank you**
>
> We would like to thank you for your comment. We are looking forward to having the opportunity of studying your findings.

---

### Author Response · Authors · 2019-11-09
**Response to all Reviewers**

This is a general response to all reviewers, meant to address common concerns and summarize our (lengthy) responses to each reviewer.

1. We would like to thank all reviewers for their excellent in-depth analysis and feedback. The resulting discussion here helps us in improving our work, in particular discussion of results and conclusions in our manuscript.

2. We would kindly ask them to carefully check again our claimed contributions in page 2. For instance, we claim neither a novel component improving AL nor a combination of components for the first time.

3. We want to stress that we take a neutral position towards AL as well as unsupervised/semi-supervised learning, and we will polish expressions like "spectacular".

4. Nevertheless, our main message, based on empirical findings, is that there are flaws, not in (deep) AL itself, but rather in its evaluation. This is clearly a negative result, however our detailed results allow for actions that will help further progress in (deep) AL. For instance:
(a) If the objective is best performance for least labels and the cost of labeling is more important than the cost of model training, it makes sense to evaluate new AL ideas in the presence of methods that use unlabeled data during model training, since this always improves performance by a large margin.
(b) One may revisit the standard protocol of equal label budget per cycle as well as the initial seed of labels, since AL is sensitive in the quality of the representation and may be outperformed by vanilla training (Random baseline) when labels are few, especially in the presence of methods that use unlabeled data during model training.
(c) If all training examples are used in each cycle, then fine-tuning rather than training from scratch in each cycle or even end-to-end training with all three components (AL, unsupervised, semi-supervised) rather than in different stages are interesting directions to explore. There are already recent approaches integrating unsupervised and semi-supervised learning (e.g. [Zhai et al., S4L: Self-Supervised Semi-Supervised Learning; ICCV19]); the same could happen with AL.

5. We believe that suggesting a new protocol or a new integrated algorithm is beyond the scope of this work. There are too many possible directions from here.

6. We also believe that our experimental setup is already complex enough, such that adding more options for the components that are orthogonal to AL would blur the presentation and discussion of results and shift it away from our main message.

7. Finally, we shall update our discussion of results and conclusion based on the above.

---

### Decision · Program_Chairs · 2019-12-19

**Decision:**

Reject

**Comment:**

This paper argues that incorporating unsupervised/semi-supervised learning into the training process can dramatically increase the performance of models. In particular, its incorporation can result in performance gains that dwarf the gains obtained by collecting data actively alone. The experiments effectively demonstrate this phenomenon.

The paper is written with a tone that implicitly assumes that "active learning for deep learning is effective" and therefore it is a surprise and a challenge to the status quo that using unlabelled data in intelligent ways alone gets such a boost. On the contrary, reviewers found that active learning not working very well for deep learning is a well-known state of affairs. This is not surprising because the most effective theoretically justifiable active learning algorithms rely on finite capacity assumptions about the model class, which deep learning disobeys.

Thus, the reviewers found the conclusions to lack novelty as the power of semi-supervised and unsupervised learning is well known. Reject.